# Eliminating Vertical Transmission of HIV in South Africa: Establishing a Baseline for the Global Alliance to End AIDS in Children

**DOI:** 10.3390/diagnostics13152563

**Published:** 2023-08-01

**Authors:** Ahmad F. Haeri Mazanderani, Tanya Y. Murray, Leigh F. Johnson, Mathilda Ntloana, Tabisa Silere-Maqetseba, Sufang Guo, Gayle G. Sherman

**Affiliations:** 1Department of Paediatrics & Child Health, Faculty of Health Sciences, University of the Witwatersrand, Johannesburg 2196, South Africa; 2Centre for HIV & STIs, National Institute for Communicable Diseases, National Health Laboratory Service, Johannesburg 2131, South Africa; 3Centre for Infectious Disease Epidemiology and Research, University of Cape Town, Cape Town 7925, South Africa; 4National Department of Health, Pretoria 0001, South Africa; 5United Nations Children’s Fund, Pretoria 0011, South Africa

**Keywords:** early infant diagnosis, vertical transmission, global alliance, paediatric HIV, end AIDS

## Abstract

To gain a detailed overview of vertical transmission in South Africa, we describe insights from the triangulation of data sources used to monitor the national HIV program. HIV PCR results from the National Health Laboratory Service (NHLS) were analysed from the National Institute of Communicable Diseases (NICD) data warehouse to describe HIV testing coverage and positivity among children <2 years old from 2017–2021. NICD data were compared and triangulated with the District Health Information System (DHIS) and the Thembisa 4.6 model. For 2021, Thembisa estimates a third of children living with HIV go undiagnosed, with NICD and DHIS data indicating low HIV testing coverage at 6 months (49%) and 18 months (33%) of age, respectively. As immunisation coverage is reported at 84% and 66% at these time points, better integration of HIV testing services within the Expanded Programme for Immunization is likely to yield improved case findings. Thembisa projects a gradual decrease in vertical transmission to 450 cases per 100,000 live births by 2030. Unless major advances and strengthening of maternal and child health services, including HIV prevention, diagnosis, and care, can be achieved, the goal to end AIDS in children by 2030 in South Africa is unlikely to be realised.

## 1. Introduction

The South African National Department of Health has committed to ‘The Global Alliance to end AIDS in children by 2030.’ This partnership aims to reinvigorate comprehensive HIV care and improve clinical outcomes in children and adolescents [1]. The work of the Global Alliance is aligned with four pillars, namely (1) Early testing and optimal treatment and care for infants, children, and adolescents (2) Closing the treatment gap for pregnant and breastfeeding women living with HIV to eliminate vertical transmission (3) Preventing new HIV infections among pregnant and breastfeeding adolescent girls and women; and (4) Addressing rights, gender equality, and the social and structural barriers that hinder access to services.

Progress in reducing vertical transmission of HIV has been remarkable since 2011, when the ‘Global Plan towards Elimination of New HIV Infections among Children and Keeping their Mothers Alive by 2015’ was launched [2]. However, criteria for the elimination of vertical transmission (EVT), which include reducing new HIV cases to fewer than 50 per 100,000 live births, have been difficult to achieve, with only a handful of countries succeeding [3]. Nevertheless, remarkable progress in reducing the vertical transmission rate has been made in South Africa, with early transmission at around 6 weeks of age declining from 25–30% in 2001 to approximately 1.4% in 2016 [4]. Whereas South Africa’s paediatric HIV incidence has in the past been reported using epidemiological surveys, routine data sources continue to be used to gauge paediatric HIV testing coverage and vertical transmission rates. These include data from the National Health Laboratory Service (NHLS), as reported from the National Institute for Communicable Diseases (NICD) data warehouse, and the District Health Information System (DHIS). Data from the NHLS represents all clinical laboratory data from South Africa’s public health sector and is considered the most accurate source of laboratory test data. Alternatively, the DHIS is a much broader source of public health data and is used to collect aggregated routine health data from over 4000 public healthcare facilities to support the planning and monitoring of health services. Additionally, the Thembisa mathematical model is widely used to monitor the country’s HIV programme [5]. The latest Thembisa 4.6 version of the model has been revised to include assumptions associated with the introduction of routine birth and 6-month testing among HIV-exposed infants.

In the following study, triangulation of South African HIV information sources was undertaken to document trends, establish a baseline for early HIV testing coverage and vertical transmission, and facilitate the identification of gaps and monitoring of interventions to contribute towards the country’s Global Alliance workplan.

## 2. Materials and Methods

### 2.1. Diagnostic Setting

The South African National HIV Guidelines’ testing recommendations are mostly aligned with the scheduled clinic visits of the Expanded Program of Immunization (EPI) to facilitate an integrated approach to comprehensive child care services. HIV Polymerase Chain Reaction (PCR) testing is recommended for all HIV-exposed infants at birth to detect intrauterine transmission, around 10-weeks to detect intrapartum transmission, at 6-months to detect early postnatal transmission, and at 6 weeks post cessation of breastfeeding to exclude postnatal transmission. Any child presenting with symptoms of HIV infection is eligible for an age-appropriate HIV test. All positive HIV PCR tests should be confirmed on a subsequent sample. The recommendation for testing at 6-months was introduced in December 2019 [6], just prior to the COVID-19 pandemic, and was informed by the South Africa Prevention of Mother-to-Child Transmission Evaluation study finding that 80% of all vertical HIV transmission in pregnant and breastfeeding women living with HIV (PBWHIV) diagnosed during antenatal care in South Africa occurs by 6-months of age [7]. These new guidelines also recommended that all children, regardless of HIV exposure, have a rapid HIV test or HIV ELISA at 18 months of age, which, if reactive, be confirmed with an HIV PCR test (on account of the potential to detect maternal antibodies during infancy and early childhood, antibody-based HIV tests are only deemed diagnostic after 24 months of age) [8]. The rationale for universal testing at 18 months of age is to identify and link to care for all vertically HIV-infected children at that time point. Reasons for HIV-infected children only being identified at 18 months include being previously diagnosed but not linked to care or lost to follow-up post linkage, never tested, or born to PBWHIV with undetected HIV infection or late seroconversion.

Essentially, all children living with HIV (CLHIV) <2 years of age should be diagnosed by HIV PCR testing, with guidelines supporting multiple testing opportunities to facilitate timely diagnosis and linkage to care during infancy and early childhood.

### 2.2. Data Sources and Age Disaggregation

The NHLS, which serves an estimated 80% of the South African population, performs laboratory diagnostic services in around 260 laboratories nationally. All these test data are stored centrally, and the NICD, a division of the NHLS, analyses the HIV test data to provide near real time HIV programme ‘data for action’ reporting designed to assist in improving patient outcomes [9]. For this analysis, the total number of HIV PCR tests and the total number of first HIV PCR positive tests performed <2 years of age were extracted. The ‘first HIV PCR positive test’ was obtained using the NHLS Corporate Data Warehouse (CDW) algorithm, which links multiple tests belonging to a single patient based on a scored probability of how closely patient demographics, viz., name, surname, and date of birth, match [10]. The earliest HIV PCR positive test per patient is the ‘first HIV PCR positive test’ which theoretically equates to a newly infected child. However, poor data quality and a linking algorithm set to maintain high specificity translate into the under-matching of tests and therefore over-counting of newly-diagnosed HIV-infected children using the first HIV PCR positive test indicator. To remedy this, a manual deduplication exercise was undertaken on 4870 HIV PCR positive tests in 2021 to estimate the percentage overestimation of first PCR positive tests at different age intervals (personal communication: GG Sherman). 70% of this deduplication exercise was on HIV PCR-positive tests from the Western Cape Province, known to have the best data quality and therefore the highest chance of detecting duplicate tests. These findings were integrated into the NICD data by reducing the number of first HIV PCR positives by 10% and 30% at ages <7 days and 7 days–<7 months, respectively. Negligible numbers of duplicate HIV PCR positive tests were detected between 7 and 24 months of age. NICD PCR positives were disaggregated by age such that birth, 10-week, and 6-month testing were defined as testing occurring at <7 days, 7 days–<3 months, and 5–7 months of age, respectively. The vast majority of total HIV tests are comprised of HIV PCR-negative results, for which no estimate of the underperformance of the CDW linking algorithm is available. Therefore, the NICD age ranges for total HIV tests performed have been narrowed to reduce the likelihood of overcounting. For example, total tests performed at birth are counted as those performed at <3 days and 10-weeks at 2–3 months of age, to reduce the chances of counting potential duplicate testing performed at the 6-day postnatal or 6-week immunisation visits, respectively.

DHIS indicators used for this analysis included the total number of HIV PCR tests performed and the number of HIV PCR positive tests at birth (0–<6 weeks of age) and around 10-weeks (6–<14 weeks of age); the total number of HIV rapid tests performed and positive results at 18 months of age; the total number of live births at a facility; the total number of live births to women living with HIV (WLHIV); and the public sector immunisation coverages at birth, 10-weeks, and 6- and 18 months of age. As indicated above, the age ranges for birth and around 10-week PCR testing differ between NICD and DHIS data. NICD defines total birth positives as <7 days, compared with DHIS at 0–<6 weeks, to further counteract inaccuracies in deduplication and achieve a more accurate estimate of neonates testing HIV PCR positive for the first time. This is based on NICD data demonstrating that 95% of the total HIV PCR tests performed between 0–<6 weeks occur at <7 days. Furthermore, by limiting the interval of the various NICD age categories, more accurate patient-level reporting can be achieved with the CDW record linking algorithm, as previously demonstrated [10]. For the years 2017–2019, HIV rapid test data was reported on DHIS. Although the 2019 HIV testing guidelines recommended that all children undergo HIV antibody testing at 18 months of age, the national indicator data set defined this indicator for HIV-exposed children only, which may have resulted in underreporting.

Thembisa 4.6 estimates used include new HIV diagnoses <1 year of age, 1–<2 years of age, and new HIV infections at/before birth and due to breastfeeding <3 years, total births, and births to HIV-positive mothers. The model assumes vertical transmission rates at birth and during breastfeeding depend on the timing of maternal HIV acquisition (with transmission rates being particularly high during the acute phase of HIV infection), the timing of maternal ART initiation, infant feeding practises (with the lower postnatal transmission in the case of exclusive breastfeeding compared to mixed feeding), and the maternal CD4 count in untreated mothers [11]. Women who are HIV-diagnosed are assumed to have shorter durations of breastfeeding than undiagnosed HIV-positive mothers and HIV-negative mothers. Infants who acquire HIV at/before birth are assumed to have high rates of HIV disease progression and mortality in the absence of antiretroviral treatment (ART), while children who acquire HIV postnatally are assumed to have relatively slow disease progression. The model allows for early infant diagnosis, testing at 18 months, and testing in children at other times (most frequently because of HIV-related symptoms). The model is calibrated to HIV prevalence data from antenatal clinic surveys, and the paediatric component of the model is further calibrated to a number of additional data sources, including paediatric HIV prevalence in household surveys, recorded deaths, total antibody tests, and testing yields, total children on ART, and the age distribution of children on ART [12]. There are a number of important differences between Thembisa estimates and DHIS and NICD data sources. For example, the number of CLHIV patients who are diagnosed according to the Thembisa model is the estimated number where positive test results are returned to the caregiver, whereas NICD and DHIS indicators are based on total positive results. More broadly, the Thembisa estimates are for the whole country, including the private healthcare sector, whereas HIV indicators from DHIS and NICD are restricted to the public sector.

NICD and DHIS monthly data were extracted for a five-year period spanning 1 July 2017 to 30 June 2022. Data were analysed such that flow variables (e.g., testing volumes and positivity) for each year were reported from the middle of the stated calendar year to the middle of the following calendar year to match the time intervals used in the Thembisa model. Output definitions, age ranges, and calculations are summarised in Table 1. Flow variables were categorised by age as <1 year referring to the first year of life, 1–<2 years referring to the second year of life, and <2 years referring to the first two years of life.

### 2.3. Testing Coverage and Positivity

NICD and DHIS testing coverages at different ages were calculated by dividing the total number of HIV PCR tests performed by the total number of live births to WLHIV—except for the 18-month age group, where the total number of live births at a facility was used as the denominator (Table 1). The percentages of children immunized at the same time intervals as HIV testing (birth, 10-weeks, 6-months, and 18-months) were obtained from DHIS for the public health sector.

The number of infants with a positive result per annum for the period 2017–2021 is described for NICD and DHIS data sources at birth and 10-weeks, with data also available from NICD at 6 months of age. Additionally, for the period 2015–2019, DHIS reported on the number of positive HIV rapid tests at 18 months of age, thereby allowing a comparison with the NICD number of positive HIV PCR results between 18–24 months of age.

Case rates of newly infected children for the Thembisa model were calculated by dividing the new HIV infections at/before birth and due to breastfeeding, accounting for all vertical transmission among CLHIV <3 years of age, by the total number of live births in the country as estimated by the model. Additionally, Thembisa estimates for new HIV infections at <2 years of age were used as an alternate numerator that would be more comparable to the NICD data.

Case rates for the NICD and DHIS data were calculated by dividing the total number of first HIV PCR positive tests within <2 years by the NICD by the DHIS total number of live births at a facility. Case rates are expressed per 100,000 live births.

Annual trends are presented, and direct comparisons between the datasets are assessed for the 12-month period of 1 July 2021–30 June 2022, unless otherwise stated.

### 2.4. Ethical Considerations

The National Institute for Communicable Diseases has approval from the Human Research Ethics Committee of the University of the Witwatersrand (M160667; M210752) to conduct communicable disease surveillance and analysis of routine laboratory data. All study methods were performed in accordance with the relevant guidelines and regulations. No patient-identifying data was extracted or used for this analysis.

## 3. Results

### 3.1. Live Births, HIV-Exposure and Immunization Uptake

Thembisa 4.6 estimates 1,131,820 total live births and 275,527 (24.3%) live births to WLHIV during 2021. These figures are similar to those reported by DHIS during the same period of 995,801 and 255,186 (25.6%), respectively, with the difference largely attributable to the contribution made by the private sector. Regarding estimates from the Thembisa model for 2030, the annual number of live births is projected to remain fairly stable at 1,116,220, but the number and proportion of live births to WLHIV are projected to drop to 182,012 (16.3%), suggesting a marked decrease in the antental HIV prevalence over the coming decade.

For the year 2021 within South Africa’s public health sector, 82% of children are reported to have been vaccinated at birth and 10-weeks of age, 84% at 6-months, and 66% at 18-months of age. These data are for immunizations among both HIV-exposed and HIV-unexposed infants and children, as the EPI indicator does not differentiate between these population groups.

### 3.2. Testing Coverage

Outputs from Thembisa 4.6 suggest that there is a considerable number and proportion of CLHIV patients who remain undiagnosed during infancy and early childhood. Among infants, the diagnostic gap was approximately 3915 (51.9%) in 2017, 2423 (44.3%) in 2021, and a projected 1644 (47.0%) for 2030.

Data from both the NICD and DHIS suggest close to complete testing coverage among HIV-exposed infants at the time of birth, increasing from 95% and 94% in 2017 to 100% and 99% in 2021, respectively (Figure 1). However, estimates of 10-week testing coverage vary between the two data sources, with NICD estimating an increase from 80% to 86% and DHIS data suggesting a much lower coverage ranging from 69% to 73% between 2017 and 2021. Subsequent to the introduction of HIV PCR testing at 6-months of age (for HIV-exposed infants) and universal HIV serology testing at 18-months (for all children irrespective of HIV-exposure status) in 2019, NICD data reports an increase in 6-month testing coverage to 49% by 2021, and DHIS data reports an increase in 18-month testing coverage to 33% by 2021. As these are the only data sources for each of these time points, it is not possible to compare estimates with each other.

### 3.3. Vertical Transmission

The Thembisa 4.6 model estimates the total number of vertically transmitted HIV infections among <3 year-olds to be 10,945 in 2017, 7830 in 2021, and 5043 in 2030. Most transmissions occur in the first 2 years of life, with only 7.5% occurring after 24 months. During 2021, the Thembisa model estimates that there were 7247 new HIV infections among children aged <2 years, of which 5468 (75.5%) occurred during infancy (Figure 2). The number and proportion of CLHIV patients who are estimated to have been newly diagnosed are 3045 (55.7%) among <1 year-olds and 4752 (65.6%) among <2 year-olds. NICD first PCR-positive test data for the same period found 4632 CLHIV were diagnosed <1 year of age and 5688 CLHIV were diagnosed <2 years of age, representing 84.7% and 78.5% of Thembisa estimates of new infections, respectively.

Regarding routes of vertical transmission for the year 2021, Thembisa estimates that of 7247 new HIV infections among infants and children <2 years of age, 2600 (35.9%) cases were on account of perinatal infection (i.e., intrauterine or intra-partum infections). Hence, there were an estimated 4647 (64.1%) postnatal infections (i.e., transmission via breastfeeding). Of the postnatal infections, 1779 (24.5%) are estimated to have occurred during the second year of life, equating to 38.3% of all postnatal infections. NICD data found that of all CLHIV patients <2 years of age, 1620 (28.5%) were diagnosed at birth (i.e., intrauterine infections), 1266 (22.3%) at 10-weeks (i.e., likely intrapartum or early postnatal infections), and 2802 (49.3%) were diagnosed thereafter (Figure 3).

Compared with Thembisa and NICD data, DHIS indicators for infants testing HIV PCR positive at birth and 10-weeks of age were markedly lower at 1049 and 1038 cases, respectively (Figure 3). This was associated with a markedly lower percentage of positivity at birth on DHIS of 0.4% compared with Thembisa of 0.6% and NICD of 0.7%.

Between 2015 and 2019, the annual number of HIV rapid test positives reported on DHIS decreased from 2685 to 1248. This was consistently higher than the number of HIV PCR positives from NICD, which increased from 222 to 275. The partial implementation of guidelines to perform confirmatory PCR testing between 18 and 24 months of age likely contributed to the reduction in this difference over time, evident from the ‘NICD first PCR positive tests 1–<2 years’ (Figure 2), converging towards the Thembisa estimates.

Between 2017 and 2021, the Thembisa model estimates that there was a decrease in the annual number of new HIV cases among infants and children <2 years of age, from 10,079 to 7247, as well as a decrease in the vertical transmission case rate from 886 to 640 per 100,000 live births (Figure 2 and Figure 4). According to the NICD HIV PCR-positive data, case rates declined from 749 to 571 per 100,000 live births between 2017 and 2021, with the curve approaching the Thembisa new HIV cases in CLHIV <2 years, consistent with an increase in diagnosed cases over time (Figure 4). Whereas there was also a decrease in the estimated annual number of CLHIV diagnosed in Thembisa, from 6174 to 4752 (Figure 2), representing a drop of 2832 (28%) cases, the model suggests that the proportion of CLHIV diagnosed increased slightly from 61.3% to 65.6%. During this same period, NICD data demonstrates that the number of CLHIV <2 years of age who were diagnosed decreased by 1241 (17.9%) cases. Whereas NICD birth and 10-week positives decreased by 262 (13.9%) and 530 (29.5%), DHIS birth and 10-week positives decreased by 478 (31.3%) and 536 (34.1%), respectively (Figure 3).

The projected Thembisa estimates for the year 2030 are 4663 new HIV infections among infants and children <2 years of age (Figure 2). An additional 380 (7.5%) new infections are expected to occur between 2–<3 years of age, representing a total vertical transmission case rate of 452 HIV infections per 100,000 live births (Figure 4).

## 4. Discussion

South Africa’s health data, especially for infants, urgently requires a universally used unique identifier. Until this is achieved, national longitudinal cohort monitoring is unattainable. Although the three sources of HIV estimates used in this comparison are distinctly different in the types and methods of data collection, their triangulation (with consideration of the caveats associated with each dataset) provides a more robust picture of the vertical transmission landscape than viewing each in isolation.

While the Thembisa model estimates that nearly half of HIV-infected infants remain undiagnosed during the first year of life, NICD and DHIS data demonstrate excellent testing coverage at birth but increasing testing gaps at 10-weeks and 6-months of age. Whereas the NICD 10-week testing coverage slowly increased between 2017 and 2021 from 80% to 86%, the 6-month testing coverage reached only 49%, reflecting implementation efforts that were likely stunted by the COVID-19 pandemic occurring three months after the release of the new guidelines in 2019. As the majority of vertical transmission cases are thought to occur by 6-months of age [7], increasing testing coverage at this time point is critical to improving early diagnostic case findings.

The lower testing coverage estimates of DHIS compared with NICD at these time points, and especially the lower numbers of infected infants reported, are likely accounted for by challenges associated with the manual nature of data collection within DHIS. Accurate reporting of DHIS HIV PCR positive tests at birth is particularly problematic since DHIS reporting at hospital-level is generally less well performed than at clinic-level and because birth PCR test results are usually only available once the newborn has already been discharged, often to a local primary healthcare facility, for postnatal follow-up. On account of the additional deduplication and restricted age ranges applied to the NICD data, the NICD estimates are likely to be conservative, further highlighting the underreporting of these DHIS HIV PCR-positive indicators.

The reported HIV rapid testing coverage of only 33% at 18 months of age for 2019 is also concerning. Like 6-month HIV PCR testing, universal HIV antibody testing for all children was introduced in the 2019 guidelines, and implementation may also have been hampered by the COVID-19 pandemic. Although the definition of this indicator at the time was for HIV-exposed children only, it is unlikely that data collection was restricted to this population. However, this may have resulted in some underreporting. Nevertheless, improved testing coverage at this time point, including confirmation of positive antibody tests with HIV PCR tests, will be essential for closing the diagnostic gap, as Thembisa estimates over a third of CLHIV patients <2 years of age remain undiagnosed.

Importantly, DHIS public-sector immunisation data suggests moderate vaccination coverage of >80% at 6 months of age. The difference between immunisation and early infant diagnosis coverage at this time point suggests a lack of comprehensive care at facilities where infants are presenting for their immunizations but are not being identified for HIV testing. This is even true at 18 months of age, when two-thirds of children are being vaccinated but only one-third are being tested for HIV. This critical gap in HIV testing should prompt facility-based investigations to improve the integration of health services at the primary care level.

A deeper understanding of transmission routes is also fostered through data triangulation. Although the Thembisa model estimates breastfeeding to be the predominant mode of vertical transmission, NICD data demonstrates persistent intra-uterine infections, which comprise more than a third of all diagnosed CLHIV <1 year of age. This highlights the need not only for preventing new HIV infections among pregnant and breastfeeding women but also the importance of early antenatal booking and comprehensive HIV testing services, as well as improved maternal virological control during the antenatal period, if South Africa is to eliminate vertical transmission.

As per World Health Organization criteria for validation of the elimination of vertical transmission, South Africa has to date achieved the impact target of Bronze tier status with ≤750 cases per 100,000 live births [13]. Although the antenatal HIV prevalence is expected to decrease, with HIV-exposed infants comprising approximately 16% of total live births by 2030 (from 25% in 2020), Thembisa estimates suggest that at best, only Silver-tier status of between 250 and 500 cases per 100,000 live births will be achieved by 2030. Hence, the target of ≤50 cases per 100,000 live births required for the elimination of vertical transmission will remain elusive unless additional interventions are introduced. One such potential intervention is enhanced access to pre-exposure prophylaxis (PrEP). A modelling study evaluating the impact of 80% PrEP coverage in South Africa between 2020 and 2030 has estimated a reduction in vertical transmission of approximately 40% [14]. Although this is very promising and will likely reduce the gap between new cases and those diagnosed, it is in itself not sufficient to bring down the case rates to the desired level. Hence, further strengthening of the healthcare system in conjunction with additional innovations in the field, such as long-acting injectable agents for both prevention and treatment, must be effected together if South Africa is to eliminate vertical transmission.

Limitations exist for all three sets of estimates. Statistical modelling is dependent on reliable input data, so estimates are only as valid and updated as the data used [15]. As the number of CLHIV patients who are diagnosed according to the Thembisa model refers to those cases where a diagnosis is received by caregivers, it is not directly comparable with NICD and DHIS data. However, as there has recently been considerable effort in the field to ensure all HIV PCR positive results are acted upon [9], it remains to be determined what proportion of CLHIV patients do not receive results. Furthermore, estimates presented in this study, such as the number and proportion of CLHIV patients who remain undiagnosed as well as case rates, are calculated using numerators and denominators that relate to different cohorts, resulting in potential inaccuracies (albeit minimal). Using routine programmatic data like the DHIS depends on every healthcare worker responsible for data collection at ±4000 facilities correctly and completely recording information and accurately collating it, resulting in likely underreporting. The laboratory data reported by NICD excludes indeterminate HIV PCR results (as does DHIS data) as well as eligible infants who never accessed testing, such as when a mother’s HIV status is unknown or in cases where there is poor health-seeking behaviour or infant death. Indeterminate results comprise approximately 15% of ‘HIV-detected’ PCR results, with nearly half of patients with an indeterminate result subsequently found to be HIV-infected [16]. The decision to exclude these cases was taken to account for the reduced positive predictive value (i.e., increased probability of false-positive results) among early infant diagnostic assays within the context of a decreasing transmission rate, but is likely to have resulted in underreporting of the true number of HIV-infected infants who have an HIV-detected PCR result. On the other hand, the lack of a unique patient identifier from birth makes reporting patient-level data challenging, with potential over-reporting of testing coverage and positivity.

Some of these limitations can be addressed. For instance, the most recent revision of the Thembisa model has incorporated programmatic updates that strengthen diagnostic estimates. This has, for example, resulted in a higher estimate of the proportion of CLHIV <2 years of age who are diagnosed. Whereas the Cost-Effectiveness of Preventing AIDS Complications (CEPAC) Pediatric model estimates 56% of CLHIV 2 years of age were diagnosed in 2018, Thembisa 4.6 estimates 61% of CLHIV <2 years of age were diagnosed in 2018, and this further increased to 66% in 2021 [17]. Analysis of laboratory data has been refined by limiting the age ranges used to describe testing time points and adjusting positivity outputs following additional deduplication (albeit by generating assumptions from a validation exercise involving only one province and one district). An investigation of the discrepancy in the number of PCR-positive tests at birth and 10-weeks of age between DHIS and NICD is required to more accurately count the number of infants diagnosed. However, until the Health Patient Registration System (HPRS) or other unique health identifier is issued at birth, efforts to accurately enumerate children diagnosed with HIV and longitudinally monitor them in care will be hampered.

## 5. Conclusions

Through triangulation of HIV estimates used to monitor vertical transmission in South Africa, the emerging picture highlights the need for better integration of immunisation and HIV testing services at primary healthcare facilities to improve timely diagnosis and case-finding. Furthermore, radical strengthening of maternal HIV services, including prevention, diagnosis, and care, must take place for South Africa to eliminate vertical transmission and ultimately end AIDS in children.

## Figures and Tables

**Figure 1 diagnostics-13-02563-f001:**
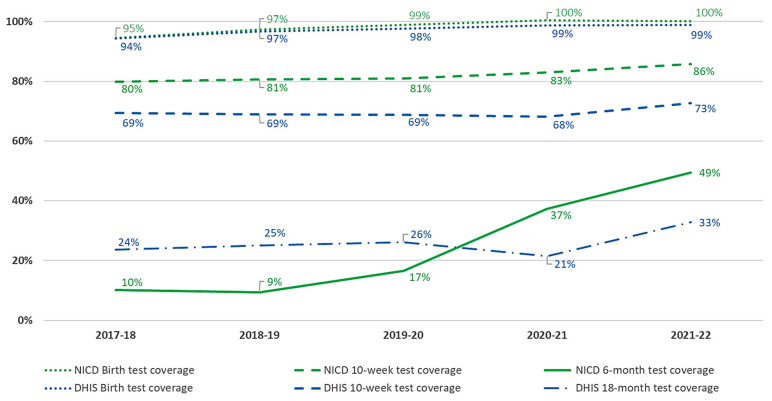
NICD and DHIS HIV PCR Test Coverage at Birth, 10-weeks, 6- and 18-months of age.

**Figure 2 diagnostics-13-02563-f002:**
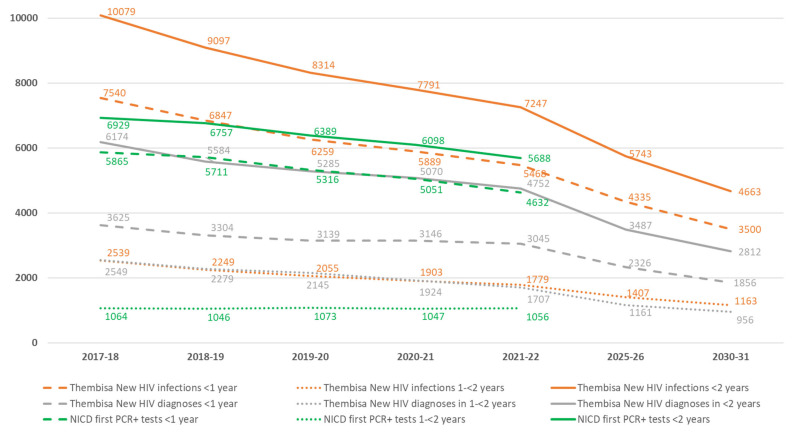
A comparison of Thembisa estimates of new HIV infections and new HIV diagnoses with NICD’s first PCR+ tests during the first two years of life.

**Figure 3 diagnostics-13-02563-f003:**
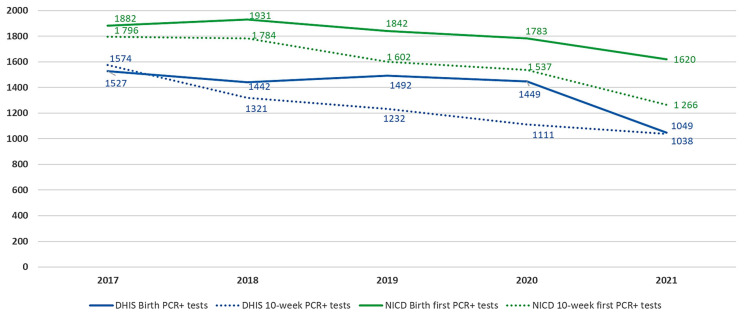
A comparison between NICD and DHIS New HIV diagnoses at birth and around 10-weeks of age.

**Figure 4 diagnostics-13-02563-f004:**
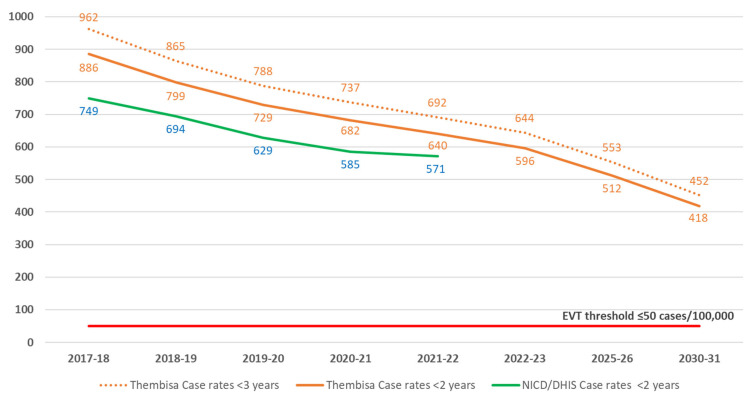
Case rates, expressed per 100,000 total live births, for Thembisa estimates at <2 and <3 years of age in comparison to NICD/DHIS case rates at <2 years of age.

**Table 1 diagnostics-13-02563-t001:** Output Definitions, Age Ranges, and Calculations.

Outputs	NICD	DHIS	Thembisa 4.6
**Total live births**		Live birth in facility	Total births
**Live births to WLHIV**		Live birth to HIV+ woman	Births to HIV+ mothers
**Total HIV infections**			New HIV infections:
			In infants (<1 y)
			In 1–<2 year olds
			At/before birth/breastfeeding (<3 y)
**Total HIV PCR tests**	Total HIV PCR tests:	Infant PCR test:	
	At birth (<3 d)	At birth (0–<6 w)	
	Around 10 weeks (2–3 m)	Around 10 weeks (6–<14 w)	
	Around 6 months (5–7 m)		
**Total HIV Rapid Tests**		HIV test around 18 months	
**Newly diagnosed CLHIV**	Total first HIV PCR+ tests:	Infant PCR test positive:	New HIV diagnoses:
	At birth (<7 d)	At birth (0–<6 w)	In infants (<1 y)
	Around 10 weeks (7 d–<3 m)	Around 10 weeks (6–<14 w)	In 1–<2 year olds
	Around 6 months (5–7 m)		
	<2 years (<1 y, 1–<2 y)		
		HIV Rapid test positive around 18 months	
**Immunisation** **Coverage**		Birth (BCG)	
		10 weeks (Hexavalent 2)	
		6 months (Measles 1)	
		18 months (Hexavalent 4)	
**Testing coverage** **calculations**	Total HIV PCR tests (NICD)/ Live birth to HIV+ woman (DHIS)	Total infant PCR tests/ Live birth to HIV+ woman	
		Total HIV (Rapid) test around 18 months/ Live birth in facility	
**Case rate calculations**	Total number of first HIV PCR positive test <2 years (NICD)/ Total Live birth in facility (DHIS)		New HIV infections at <3 years/ Total births
			New HIV infections at <2 years/ Total births

NICD, National Institute for Communicable Diseases; DHIS, District Health Information System; CLHIV, Children living with HIV; WLHIV, Women living with HIV; PCR, polymerase chain reaction; +, positive; d, days; w, weeks; m, months; y, years; blank, no data available; BCG, Bacille Calmette-Guérin vaccine for Tuberculosis; Hexavalent 2 and 4, the second and fourth doses of DTaP-IPV-Hib-HepB (diphtheria-tetanus-acellular pertussis-injectable polio-Haemophilus influenza b-Hepatitis B vaccine); Measles 1, first dose of measles vaccine. The standardized names for the indicators from DHIS and Thembisa 4.6 have been used.

## Data Availability

Thembisa model outputs are available online (https://www.thembisa.org/downloads (accessed on 1 June 2023)). Requests for laboratory data can be made via the National Health Laboratory Service Academic Affairs and Research Management System (https://aarms.nhls.ac.za/NHLS_AARMS/Public/Default.aspx (accessed on 1 June 2023)). District Health Information System can be made through the National Department of Health.

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
