# Peer review of "Eliminating Vertical Transmission of HIV in South Africa: Establishing a Baseline for the Global Alliance to End AIDS in Children"

_diagnostics, 2023, doi:10.3390/diagnostics13152563_

Round 1

Reviewer 1 Report

The spread of HIV has constantly led to a significant loss of morbidity and mortality. In this work, the authors document trends through triangulation of HIV information sources in South Africa to establish a baseline for early HIV testing coverage and vertical transmission. Further, strengthening maternal HIV services, including prevention, diagnosis, and care, must take place for South Africa to eliminate vertical transmission and ultimately end AIDS in children. Specific comments and questions are listed below to improve the quality of the manuscript.

1. We know that PCR will have false positive results, can such a situation be detected in time, and if so, how can it be removed from the statistical results?

2. Page 6, lines 221-224: A significant number of people living with HIV are not diagnosed in a timely manner. What are the reasons for this?

3. The Thembisa model was used in the article to detect HIV infection in infants from birth to 2 years old. I'm not sure how to follow the same baby from birth to 2 years old.

4. What are the reasons for the gap between immunization and HIV testing?

Typos errors need to be revised, such as in the abstract Immunisation".

Author Response

  1. We know that PCR will have false positive results, can such a situation be detected in time, and if so, how can it be removed from the statistical results?

Thank you for raising this important and relevant issue. As the vertical transmission rate declines, so too will the proportion of positive test results that are true-positives (i.e. the positive predictive value of a diagnostic assay will also decrease). Extensive work regarding ensuring accurate EID services within such a context has been conducted within South Africa, with work contributing towards WHO recommendations regarding the differentiation of clearly positive HIV PCR results and potentially false-positive results – referred to as indeterminate results (Luo R, et al. J Acquir Immune Defic Syndr 2019;82:281-6). For the purposes of this paper, only HIV PCR results that were verified as positive were included in the analysis. Hence all indeterminate results were excluded. As reports from South Africa have suggested that as many as half of all infants who test indeterminate may be HIV-infected (Radebe L, et al. CMI 2022;609:e7-13), excluding these results may have resulted in a lower number of HIV positive infants and children <2 years of age being reported by the laboratory. Acknowledgement of this point is now included in the limitations section of the manuscript as follows - Indeterminate results comprise approximately 15% of ‘HIV-detected’ PCR results, with nearly half of patients with an indeterminate result subsequently found to be HIV-infected [15]. The decision to exclude these cases was taken to account for the reduced positive predictive value (i.e. increased probability of false-positive result) among early infant diagnostic assays within the context of a decreasing transmission rate, but is likely to have resulted in underreporting of the true number of HIV-infected infants who have an HIV-detected PCR result.

  1. Page 6, lines 221-224: A significant number of people living with HIV are not diagnosed in a timely manner. What are the reasons for this?

Outputs from Thembisa 4.6 suggest that there is a considerable number and proportion of CLHIV who remain undiagnosed during infancy and early childhood – This discrepancy could arise on account of a number of programmatic considerations as well as specific definitions used and potential inaccuracies regarding the model outputs (i.e. an overestimate of the number of vertical transmission cases and/or an underestimate in the number of diagnosed infants).

Regarding the former, possible reasons include low testing coverage at 10-weeks, 6-months and 18-months of age as outlined in paragraphs 2 and 4 of the Discussion section. Reducing incident maternal infections during the postnatal period is also key to ensuring timely diagnosis. Specific mention of this is now included in paragraph 7 of the Discussion Section - One such potential intervention is the enhanced access to pre-exposure prophylaxis (PrEP). A modelling study evaluating the impact of 80% PrEP coverage in South Africa between 2020 and 2030 has estimated a reduction in vertical transmission of approximately 40% [13]. Although this is very promising, and will likely reduce the gap between new cases and those diagnosed, it is in itself not sufficient to bring down the case rates to the desired level.

In terms of some of the Thembisa model definitions and possible sources of inaccuracies, these are acknowledged in the limitations section of the Discussion - As the number of CLHIV who are diagnosed according to the Thembisa model refers to those cases where a diagnosis is received by caregivers, it is not directly comparable with NICD and DHIS data. However, as there has recently been considerable effort in the field to ensure all HIV PCR positive results are acted upon [9], it remains to be determined what proportion of diagnosed CLHIV don’t receive results. Furthermore, estimates presented in this study, such as the number and proportion of CLHIV who remain undiagnosed as well as case rates, are calculated using numerators and denominators that relate to different cohorts resulting in potential inaccuracies (albeit minimal).

  1. The Thembisa model was used in the article to detect HIV infection in infants from birth to 2 years old. I'm not sure how to follow the same baby from birth to 2 years old.

Thembisa is a mathematical model that depends on assumptions about vertical transmission rates and rates of HIV testing. Although the model is calibrated to match available testing data, it is not drawing on longitudinal data from cohort studies, i.e. there are no direct data following infants from birth to 2 years that go into the model. Rather the model estimates are extrapolations from the available testing data, conditional upon assumptions about rates of vertical transmission and maternal HIV acquisition during breastfeeding.

  1. What are the reasons for the gap between immunization and HIV testing?

This is highlighted in paragraph 5 of the Discussion Section as follows - Importantly, DHIS public-sector immunization data suggests moderate vaccination coverages of >80% at 6-months of age. The difference between immunization and early infant diagnosis coverage at this time point suggests a lack of comprehensive care at facilities whereby infants are presenting for their immunizations but are not being identified for HIV testing. This is even true at 18-months of age where two-thirds of children are being vaccinated but only one-third are being tested for HIV. This critical gap in HIV testing should prompt facility-based investigations for improving the integration of health services at primary care level.

Comments on the Quality of English Language

Typos errors need to be revised, such as in the abstract “Immunisation".

This has been revised throughout the manuscript.

Reviewer 2 Report

The authors state that NICD and DHIS indicators are based on public sector data while Thembsia estimates are for the whole country. Then the obvious question is how does the Thembsia model gets its data and why is this "total" data not accessible to state sector to improve their data sources such as NCID or DHIS? If this model is for the whole country, then why cannot it's data sources  fully replace NICD and DHIS based estimates which is part of the picture?

From Table 1 it is apparent that the model does not consider PCR tests or rapid HIV tests. So how does it identify (which data sources are used) new HIV infections in children less than 1 year and those between 1 - 2 years?

Perhaps a better way to put this is that Thembsia is not a data collection platform, but a model that uses data from platforms like NICD and DHIS (and other sources beyond the timelines covered by NCID/DHIS) to make predictions. If the accuracy of this data improves then the predictions are closer to reality. In that sense I do not understand why the model is compared to NICD and DHIS in table 1, because they are not comparable.

The explanation of figure 2 is confusing although the figure itself is a good one. I think the explanation in text could be limited to the solid lines (the total estimations / diagnosis for children < 2 years) and for the split of values between the first and second years of life, the figure should be referred to rather than trying to reexplain it.

There is a lot of confusion in numbers because authors give two values seperated as <1 year and <2 years. The latter value seemingly refers to all infections in first two years of life to the reader, but I think the authors actually mean only those ocurring in between 1-2 years in most of the situations, by this "<2 years" number. If that is the case, for clarity please stick to the following format. "XX infections ocurred during the first year of life while yy additional infections ocurred in the second year of life leading to a total of (xx+yy) infections in the < 2 -year olds".

The authors do not comment on which source of estimations can be trusted more. In lack of a gold standard (a unique identifier and a common data base to capture both public and private sector data), I wonder if triangulation is a good idea as it can amplify errors in each of these platforms when combined. Instead choosing the most accurate and trustworthy data source from the current sitatuation and deriving estimates from these could be a better idea - I would appreciate the authors thoughts on this. To me the most accurate seems to be NCID (at least in first year of life where a majority of infections are detected) since authors do not explain how Thembsia gets its data and the assumptions it makes in the modelling. 

Please see comments above

Author Response

The authors state that NICD and DHIS indicators are based on public sector data while Thembsia estimates are for the whole country. Then the obvious question is how does the Thembsia model gets its data and why is this "total" data not accessible to state sector to improve their data sources such as NICD or DHIS? If this model is for the whole country, then why cannot it's data sources fully replace NICD and DHIS based estimates which is part of the picture?

Thembisa is a mathematical model rather than a data source; it draws together different data sources and makes assumptions to “fill in” where data are missing. In the case of the private sector, we have no data on vertical transmission, so the Thembisa model makes the simplifying assumption that rates of early infant diagnosis are the same in the private sector as in the public sector. This assumption may be incorrect (as with many of the other assumptions in the model), so we would caution against the interpretation that the model estimates “replace” the estimates from other sources. In any event, the private sector itself accounts for a very small proportion of the HIV programme, and is deemed to have a minimal impact on testing coverages and case rates as reported from the public sector by NICD. As HIV is not a notifiable medical condition, the same data sharing agreements between the private sector (in which multiple different pathology laboratories operate in contrast to the single pathology provider in the public sector viz. NHLS) and NICD, as for other infectious diseases, are not operationalised.

From Table 1 it is apparent that the model does not consider PCR tests or rapid HIV tests. So how does it identify (which data sources are used) new HIV infections in children less than 1 year and those between 1 - 2 years? Perhaps a better way to put this is that Thembisa is not a data collection platform, but a model that uses data from platforms like NICD and DHIS (and other sources beyond the timelines covered by NICD/DHIS) to make predictions. If the accuracy of this data improves then the predictions are closer to reality. In that sense I do not understand why the model is compared to NICD and DHIS in table 1, because they are not comparable.

The Thembisa model uses numerous data sources and assumptions to provide outputs for a number of indicators. As stated in Paragraph 3 of the Methods Section - The model is calibrated to HIV prevalence data from antenatal clinics surveys, and the paediatric component of the model is further calibrated to a number of additional data sources, including paediatric HIV prevalence in household surveys, recorded deaths, total antibody tests and testing yields, total children on antiretroviral therapy (ART) and the age distribution of children on ART. Furthermore, the latest Thembisa 4.6 version of the model has been revised to include assumptions associated with the introduction of routine birth and 6-month testing among HIV-exposed infants. Whereas certain data from NICD and DHIS are directly comparable with Thembisa outputs, the Thembisa model does not have outputs for all indicators reported from NICD and DHIS such as HIV PCR test volumes. Additionally, for those indicators where outputs are available from NICD, DHIS and Thembisa, it is important to triangulate and account for these differences, as highlighted in the Discussion Section, to improve the accuracy of monitoring and reporting.

To avoid the potential confusion highlighted by the reviewer, the title of Table 1 has been revised to, ‘Output Definitions, Age Ranges and Calculations.’

The explanation of figure 2 is confusing although the figure itself is a good one. I think the explanation in text could be limited to the solid lines (the total estimations / diagnosis for children < 2 years) and for the split of values between the first and second years of life, the figure should be referred to rather than trying to reexplain it.

The explanatory text of Fig 2 focuses on a few points,

  • Thembisa models estimates 75,5% of new infection among CLHIV <2yrs of age occur during the first year of life
  • Whereas Thembisa estimates that only 55,7% of CLHIV infected <1yr of age are diagnosed <1yr of age, the NICD data suggests a much higher proportion are diagnosed (84,7%)

The authors believe these points are worth highlighting, and have limited the explanatory text of this figure to 6 lines.

There is a lot of confusion in numbers because authors give two values seperated as <1 year and <2 years. The latter value seemingly refers to all infections in first two years of life to the reader, but I think the authors actually mean only those occurring in between 1-2 years in most of the situations, by this "<2 years" number. If that is the case, for clarity please stick to the following format. "XX infections ocurred during the first year of life while yy additional infections ocurred in the second year of life leading to a total of (xx+yy) infections in the < 2 -year olds".

Thank you for this comment. This has been clarified in the Methods Section as follows - Flow variables are categorized by age as <1 year referring to the first year of life, 1-<2 years referring to the second year of life; and <2 years referring to the first two years of life.

The authors do not comment on which source of estimations can be trusted more. In lack of a gold standard (a unique identifier and a common data base to capture both public and private sector data), I wonder if triangulation is a good idea as it can amplify errors in each of these platforms when combined. Instead choosing the most accurate and trustworthy data source from the current situation and deriving estimates from these could be a better idea - I would appreciate the authors thoughts on this. To me the most accurate seems to be NICD (at least in first year of life where a majority of infections are detected) since authors do not explain how Thembsia gets its data and the assumptions it makes in the modelling. 

The purpose of triangulation is not to combine but rather to compare and understand why differences occur. There are some sources that might be considered more reliable for one purpose, but other sources might be considered more reliable for another purpose. We have been careful to present a balanced perspective on the strengths and limitations of the different data sources. However, it is worth noting that the NICD estimates in the first year of life might not be a gold standard because of the potential for double-counting of positive results in the same infant (a point made in Discussion section - the lack of a unique patient identifier from birth makes reporting patient-level data challenging with potential over-reporting of testing coverage and positivity). The description of the Thembisa data sources is already included in the third paragraph of section 2.1, but we have added more detail regarding the Thembisa assumptions in the Methods Section - The model assumes vertical transmission rates at birth and during breastfeeding depend on the timing of maternal HIV acquisition (with transmission rates being particularly high during the acute phase of HIV infection), the timing of maternal ART initiation, infant feeding practices (with lower postnatal transmission in the case of exclusive breast-feeding compared to mixed feeding) and the maternal CD4 count in untreated mothers [11]. Women who are HIV-diagnosed are assumed to have shorter durations of breast-feeding than undiagnosed HIV-positive mothers and HIV-negative mothers. Infants who acquire HIV at/before birth are assumed to have high rates of HIV disease progression and mortality in the absence of antiretroviral treatment (ART), while children who acquire HIV postnatally are assumed to have relatively slow disease progression. The model allows for early infant diagnosis, testing at 18 months, and testing in children at other times (most frequently because of HIV-related symptoms).

Round 2

Reviewer 1 Report

No comments.

Reviewer 2 Report

No further comments